# Digestibility and Palatability of the Diet and Intestinal Functionality of Dogs Fed a Blend of Yeast Cell Wall and Oregano Essential Oil

**DOI:** 10.3390/ani13152527

**Published:** 2023-08-05

**Authors:** Nayara Mota Miranda Soares, Taís Silvino Bastos, Gislaine Cristina Bill Kaelle, Renata Bacila Morais dos Santos de Souza, Simone Gisele de Oliveira, Ananda Portella Félix

**Affiliations:** Department of Animal Science, Federal University of Paraná, R. dos Funcionários, 1540, Curitiba 80035-050, Brazil; tais.sbastoss@gmail.com (T.S.B.); gislainecbill@gmail.com (G.C.B.K.); renata.bacila@ufpr.br (R.B.M.d.S.d.S.); sgoliveira@ufpr.br (S.G.d.O.); apfelix@ufpr.br (A.P.F.)

**Keywords:** feed additives, food preference, intestinal fermentation products, microbiota

## Abstract

**Simple Summary:**

Prebiotics and essential oils are some of the feed additives that can be added to pet food, individually or in combination. These additives can exert important properties on the animal, such as antioxidant, anti-inflammatory, and immunomodulatory effects, as well as modulation of gut microbiota. Therefore, the aim of this study is to evaluate the combined effects of yeast cell wall and oregano essential oil on nutrient digestibility, diet palatability, intestinal fermentation products, and fecal microbiota in dogs. The addition of the blend with yeast cell wall and oregano essential oil to the dog’s diet reduced the apparent total tract digestibility of dry matter and the intake ratio compared to the control diet, modulated fecal microorganisms considered beneficial, caused greater bacterial diversity, and lowered histamine, phenol, and ammonia concentrations.

**Abstract:**

Feed additives, such as prebiotics and essential oils, are used in pet foods and can affect digestibility, palatability, and intestinal functionality of dogs. The combined effects of yeast cell wall and oregano essential oil on apparent total tract digestibility (ATTD) and palatability of diet, intestinal fermentation products, and fecal microbiota in dogs were analyzed. Eighteen adult dogs were fed for 20 days with three dry extruded diets for adult dogs: control (without the additive), a diet containing 1.5 kg/ton of yeast cell wall and oregano essential oil (1.5YCO), and a diet containing 3.0 kg/ton of yeast cell wall and oregano essential oil (3.0YCO). The inclusion of both levels of YCO reduced the intake ratio. The addition of 3.0YCO reduced the ATTD of dry matter, compared to the control group (*p* < 0.05). There were greater putrescine and cadaverine concentrations and lower histamine and ammonia (*p* < 0.05) in the feces of dogs fed 3.0YCO. In addition, fecal odor of dogs fed YCO was less fetid than the control group (*p* < 0.05). There was greater fecal bacterial diversity in dogs fed with both dietary concentrations of YCO evaluated (*p* < 0.05). Dogs fed 1.5YCO and 3.0YCO showed higher relative abundance of *Blautia* and *Faecalibacterium* and lower abundance of *Streptococcus* (*p* < 0.05) in the feces, in comparison to the control group. Given the modulation of microorganisms considered beneficial and the lower fecal concentrations of histamine, phenols, and ammonia, the YCO blend resulted in indicators of improvement of intestinal functionality in dogs.

## 1. Introduction

Maintaining a good gut functionality generally depends on diet composition and keeping a well-balanced gut microbiota [1]. In this context, many commercial foods already include prebiotics, which are ingredients that aim to improve intestinal functionality by modulating the gut microbiota [2,3]

According to the International Scientific Association of Probiotics and Prebiotics (ISAPP), a prebiotic is defined as “a substrate that is selectively utilized by host microorganisms conferring a health benefit” [4]. In other words, prebiotics are selectively fermented by the enzymes of unicellular organisms, leading to changes in the population and/or activity of the gut microbiota and providing an enabling environment for the growth of bacteria considered beneficial [1,3,5,6]. Prebiotics can be extracted from natural sources, such as plants or yeast, or can be produced through acid or enzymatic hydrolysis of polysaccharides [5]. One of the prebiotics widely used in the nutrition of dogs and cats is the mannan oligosaccharide (MOS), a short-chain carbohydrate composed of 3 to 10 mannose residues, capable of modulating the intestinal microbiota. MOS is usually obtained through enzymatic, alkaline, or acidic hydrolysis of the cell wall mannans derived from yeast (*Saccharomyces cerevisiae*) cell walls or from plant galactomannans [7].

β-glucans can also be obtained from *S. cerevisiae* cell walls. These are a group of glucose polymers that escape digestion and alter the composition of the gut microbiota, promoting mainly immunomodulatory effects in dogs [8]. β-glucans are classified as natural immunostimulants or biological response modifiers, since they interfere with various cell types and biological processes in dogs, such as effects on immune cells and the regulation of stress or cholesterol levels [9].

In addition to prebiotics, other natural additives, such as essential oils (EOs), can contribute to gut functionality and to diet digestibility and palatability in dogs. EOs are natural bioactive compounds extracted from plant parts, such as leaves, flowers, seeds, and roots. They are volatile at room temperature, aromatic, and liquid, and, in addition to promoting gut benefits, some EOs have antimicrobial, anti-inflammatory, and antioxidant potential, especially observed in pigs and broilers [10]. An example is the oregano EO, obtained by drying the leaves and flowers of *Origanum vulgare*. This EO presents the phenolic isomers carvacrol and thymol as the main components, which constitute about 78–85% of the oil [11]. These bioactive compounds are strongly related to the important antimicrobial (especially against Gram-positive bacteria) and antioxidant properties of oregano EO [12].

In addition to intestinal functionality, digestibility and palatability of the diet can be influenced by the inclusion of feed additives [1,13,14]. Even though yeast products are known as palatability enhancers in dogs [15], research about the effects of EOs on diet palatability and digestibility are limited. As such, it is possible that the combination of yeast cell wall products and oregano EO has additional effects on intestinal functionality and on diet digestibility and palatability in dogs.

To the author’s knowledge, no studies on the combination of prebiotics and EOs in canine nutrition has been published. Therefore, the aim of this study was to evaluate the combined effects of yeast cell wall and oregano EO on nutrient digestibility, diet palatability, intestinal fermentation products, and fecal microbiota in dogs.

## 2. Materials and Methods

The experiment was approved by the Ethics Committee on Animal Use of the Agricultural Sciences Sector of the Federal University of Paraná under protocol number 022/2020.

### 2.1. Experiment I: Digestibility, Fecal Characteristics, Intestinal Fermentation Products, and Microbiota

#### 2.1.1. Animals and Housing

The experiment was performed at the Research Laboratory in Canine Nutrition (LENUCAN) at the Federal University of Paraná, Curitiba, Brazil. Eighteen adult dogs (sixteen beagles, one whippet, and one mixed breed) were used; of these, there were eight males and ten females, with an average age 4.5 years old and a mean body weight of 13.3 ± 1.07 kg. The animals were clinically healthy and were individually housed in covered masonry kennels with solarium (5 m long × 2 m wide), as usual. The dogs had access to an outdoor area (around 1300 m^2^) during most of the experimental period, except during the period of fecal collection, in which they were kept only in the stalls, with access to bedding and water. The facilities had side walls with bars, which allowed limited interaction with other animals and humans.

The dogs were fed for 20 days, twice a day (7:30 a.m. and 3:30 p.m.), in an amount sufficient to meet their energy requirements for maintenance, according to the equation proposed by the NRC [16]: 130 kcal × body weight (kg)^0.75^. Water was provided *ad libitum.* Feed intake was measured daily, and body weight was recorded weekly.

#### 2.1.2. Experimental Diets

Three diets were evaluated: control (without the additive), a diet containing 1.5 kg/ton of yeast cell wall and oregano EO (1.5YCO), and diet containing 3.0 kg/ton of yeast cell wall and oregano EO (3.0YCO) (Advanced Pet Biobalance FT, Alltech, Nicholasville, KY, USA). The analyzed chemical composition of the diets is described in Table 1. The minimum concentration of MOS in the product was 25 g/kg and β-glucans was 5.83 g/kg. Calcium carbonate was used as a vehicle. The product was included by coating, together with oil and liquid palatant, after the diet extrusion process. The basal diet of the experiment was a commercial complete food formulated to meet the nutritional needs of adult dogs according to the European Pet Food Industry Federation (FEDIAF) [17]. The diet did not contain any feed additive that could influence gut functionality.

Ingredient composition: poultry by-product meal, meat meal, corn, soybean meal, poultry fat, pork liver hydrolysate, sodium chloride, citric acid, antioxidants (BHT, BHA), propionic acid, vitamin A, vitamin D3, vitamin E, vitamin B1, vitamin B6, vitamin B12, vitamin K3, nicotinic acid, folic acid, biotin, calcium pantothenate, zinc sulfate, calcium iodate, sodium selenite, copper sulfate, iron sulfate, manganese sulfate, and zinc oxide.

#### 2.1.3. Experimental Design

The dogs were subdivided into three groups according to sex, body condition score, and body weight. The control group consisted of two males and four females (including non-beagle dogs), and the 1.5YCO and 3.0YCO group included three males and three females each. Each group received an experimental diet in a completely randomized design for 20 days, totaling six replicates per treatment (each replicate corresponds to one animal). Of the total of 20 days, 15 days corresponded to the adaptation period to the diet and 5 days to the fecal collection. On the last day of the trial, fresh feces were collected.

#### 2.1.4. Digestibility and Metabolizable Energy Determination

The digestibility assay was performed based on the total fecal collection method recommended by the Association of American Feed Control Officials (AAFCO) [18]. The diets were provided for an adaptation period of 15 days, followed by 5 days of total fecal collection, totalizing 20 days of feed intake.

During the last five experimental days, the feces were collected and weighed at least twice a day for digestibility analysis. Afterward, they were placed in individual plastic containers, previously identified, covered, and stored in a freezer for further analysis. At the end of each collection period, feces were thawed at room temperature and homogenized.

Feces were dried in a forced-ventilation oven (320-SE, Fanem, São Paulo, Brazil) at 55 °C for 48–72 h or until a constant weight was reached. After drying, feces and diet samples were ground using a Willey hammer mill (Arthur H. Thomas Co., Philadelphia, PA, USA) with 1 mm sieves and subjected to chemical analysis. The analyses for diets and feces were as follows: dry matter at 105 °C (DM105), acid hydrolyzed ether extract (AEE method 954.02), ash (method 942.05), crude fiber (CF, method 962.10), calcium (method 927.02), phosphorus (method 984.27), nitrogen (N, method 954.01), and crude protein (CP), which was calculated as N × 6.25, according to the Association of Official Analytical Chemists (AOAC) [19]. Gross energy (GE) was determined using a bomb calorimeter (Parr Instrument Co., model 1261, Moline, IL, USA), and organic matter (OM) was established as 100% DM—% ash.

#### 2.1.5. Fecal Characteristics, Intestinal Fermentation Products, and Microbiota

Fecal characteristics were evaluated by total DM content (DMf), fecal output, fecal score, ammonia, odor, and pH (measured in 2 g of stool diluted in 20 mL of distilled water).

Fecal ammonia was analyzed according to Brito et al. [20], in which 5 g of fresh feces were incubated in a 500 mL lidded glass balloon, including 250 mL distilled water, for 1 h. In sequence, three drops of octyl alcohol (1-octanol) and 2 g of magnesium oxide were added to the solution, subsequently distilled in a macro-Kjeldahl apparatus and recovered in a beaker, with 50 mL boric acid. Ammonia was titrated using standardized sulfuric acid at 0.1 N.

Fecal output was calculated as g feces/g DM intake/day, and DMf was estimated considering the following: (DM at 55 °C × DM at 105 °C)/100. The fecal score was measured considering grades from 1 to 5, as follows: 1 = watery (liquid that can be poured); 2 = soft and unformed; 3 = soft, formed, and moist; 4 = hard, formed, and dry stool; 5 = hard and dry pellets [21].

For the evaluation of fecal odor, fresh feces were sampled on the last day of collection at 08:30 am. Feces were randomly collected from three animals per treatment. The samples were homogenized by treatment and placed in equal amounts (40 g) in jars of the same size and volume, which were covered with plastic film containing the same number and size of holes, to preserve odor. The evaluation was carried out by 17 people, by comparing the fecal odor of dogs fed the control diet with the feces of dogs fed the 1.5YCO and 3.0YCO diets. Values were assigned in relation to the odor of the control diet, as follows: 1—better than the control diet; 2—same as the control diet; 3—worse than the control diet. All participants smelt the odor of the standard stool and subsequently the odor of the test samples. The test samples were identified as ‘A’ and ‘B’, in order to ensure the anonymity of the treatments to the participants.

On the 20th experimental day, fresh feces were collected from all animals up to 15 min after defecation and analyzed for short-chain (SCFA) and branched-chain fatty acids (BCFAs), pH, ammonia, biogenic amines, and microbiota. For the analysis of SCFAs and BCFAs, 10 g of stool sample was weighed and mixed with 30 mL of 16% formic acid. This mixture was homogenized and stored in a refrigerator at 4 °C for a period of 3 to 5 days. Subsequently, these solutions were centrifuged at 5000× *g* in a centrifuge (2K15, Sigma, Osterodeam Hans, Germany) for 15 min. At the end of centrifugation, the supernatant was separated and centrifuged again. Each sample went through three centrifugations, and at the end of the last one, part of the supernatant was transferred to a properly labeled microcentrifuge tube for further freezing. Afterward, the samples were thawed and centrifuged again at 14,000× *g* for 15 min (Rotanta 460 Robotic, Hettich, Tuttlingen, Germany). Fecal SCFAs were analyzed by gas chromatography (SHIMADZU, model GC-2014, Kyoto, Japan). A glass column (Agilent Technologies, HP INNO wax—19091N, Santa Clara, CA, USA) 30 m long and 0.32 mm wide was used. Nitrogen was the carrier gas, with a flow rate of 3.18 mL/min. The working temperatures were 200 °C at injection, 240 °C on the column (at the rate of 20 °C/min), and 250 °C at the flame ionization detector.

Biogenic amines were analyzed according to previous procedure [22], with 0.5 g of fresh feces collected within 30 min of defecation, which were stored in 7 mL of 5% trichloroacetic acid and refrigerated. In sequence, the samples were centrifuged at 10,000× *g* for 20 min at 4 °C, and the supernatant was filtered. The biogenic amines were separated by ion-paired liquid chromatography.

For the determination of the fecal microbiota, approximately 2 g of fresh feces were placed in a sterile microcentrifuge tube and stored in a −80 °C freezer until the analysis. The commercial kit “ZR Fecal DNA MiniPrep^®^” from Zymo Research (Zymo Research, Irvine, CA, USA) was used to extract DNA from the samples, following the manufacturer’s recommended protocol. The extracted DNA was quantified by spectrophotometry at 260 nm. To evaluate the integrity of the extracted DNA, all samples were run by 1% agarose gel electrophoresis. A 250-base segment of the V4 hypervariable region of the 16S ribosomal rRNA gene was amplified using the universal primers 515F and 806R and the following PCR conditions: 94 °C for 3 min, 18 cycles of 94 °C for 45 s, 50 °C for 30 s, and 68 °C for 60 s, followed by 72 °C for 10 min. From these amplifications, the metagenomic library was built using the commercial kit “Nextera DNA Library Preparation Kit” from Illumina^®^. The amplifications were pooled and subsequently sequenced on the Illumina^®^ “MiSeq” sequencer [23]. The reads obtained on the sequencer were analyzed on the QIIME (Quantitative Insights Into Microbial Ecology) platform [24], following a workflow for removing low-quality sequences, filtering, removing chimeras, and taxonomic classification. Sequences were classified into bacterial genera by recognizing operational taxonomic units (OTUs); in this case, the homology between sequences when compared to a database. The 2017 update (SILVA 128) of the ribosomal sequence SILVA database [25] was used to compare the sequences. To generate the classification of bacterial communities by OTU identification, 611 reads per sample were used, in order to normalize the data and not compare samples with different numbers of reads, thus, avoiding bias in taxonomy. Metagenomic profiles were analyzed in STAMP software (STAMP v2.1.3) [26].

#### 2.1.6. Calculations and Statistical Analysis

Based on laboratory results, the apparent total tract digestibility (ATTD) of the nutrients and the ME of the diets were calculated, according to AAFCO [18]. For this, the following was considered:ATTD = (g nutrient intake − g nutrient excreted)/g nutrient intake(1)
ME (MJ/kg) = {kJ/g GE intake − kJ/g fecal GE − [(g CP intake − g fecal CP) × (5.23 kJ/g)]}/g feed intake.(2)

Data were analyzed according to a completely randomized design, with a total of six experimental units per treatment. Data were analyzed using the SAS statistical package (version 8, SAS Institute Inc., Cary, NC, USA). Data were analyzed for normality by the Shapiro–Wilk test and when this assumption was met, they were subjected to analysis of variance (ANOVA), considering *p* < 0.05 as significant for the F test. Means were compared by Tukey’s test (*p* < 0.05). Non-parametric data were analyzed using the Kruskal–Wallis test (*p* < 0.05).

Microbial diversity was estimated by Shannon and Chao1 indexes. Beta-diversity was analyzed by principal coordinate analysis (PCoA) using the Bray–Curtis dissimilarity method. Analysis of similarity (ANOSIM) was used to compare the overall microbiome profile among the groups considering *p* < 0.05.

### 2.2. Experiment II: Palatability Assay

#### 2.2.1. Animals and Experimental Procedures

The experiment was conducted in the Research Laboratory in Canine Nutrition (LENUCAN) of the Federal University of Paraná, Curitiba, Brazil, under the same conditions previously described. Sixteen adult Beagle dogs (eight males and eight females), 4.3 years old on average and with a mean body weight of 11.3 ± 1.07 kg, were used.

The palatability trial was conducted after the last day of Experiment I, for two consecutive days, once a day, to evaluate the palatability of YCO compared to the control diet and between the different concentrations of the additive. Three comparisons were made: control vs. 1.5YCO (Test 1), control vs. 3.0YCO (Test 2), and 1.5YCO vs. 3.0YCO (Test 3). All animals in this experiment were used in the three comparisons. Each test was conducted over two consecutive days, for a total of six days of palatability tests.

Each animal was offered two feeders simultaneously, each containing an experimental diet. At each feeding, the animals received the daily energy requirement plus 30% of each diet (average addition of 90 g), based on the NRC [16] recommendation for adult dogs in maintenance, thus, ensuring the presence of leftovers. The food was available to the animals for 30 min or until they fully consumed one of the foods. On the second day of each test, the position of the feeders was alternated to avoid laterality.

The first feeder that the animal approached during the simultaneous offering of the diets was recorded as the first choice. The diet intake ratio (IR) was calculated using the following equation:IR = [g consumed of A or B/g total consumed (A + B)] × 100.

#### 2.2.2. Statistical Analysis

The IR results were compared by paired Student’s *t*-test at 5% significance and the first choice by Chi-square test at 5%, totaling 32 replicates per test (two days × 16 dogs).

## 3. Results

### 3.1. Experiment I: Digestibility Assay and Fecal Characteristics, Intestinal Fermentation Products, and Microbiota

The ATTD of the diets and fecal characteristics of dogs are presented in Table 2. There were no episodes of vomiting or diarrhea, and all dogs consumed the diets normally, without difference among treatments (average = 198.9 ± 43.91 g DM intake/dog/day, *p* > 0.05).

Significantly lower ATTD of DM was observed with the inclusion of 3.0YCO compared to the control diet (*p* < 0.05). However, the ATTD of other nutrients did not differ among diets (*p* > 0.05). Regarding fecal characteristics, there was a lower concentration of ammonia in the feces of dogs fed 3.0YCO as compared to both other diets. The other fecal characteristics, such as pH, DMf, fecal score, and fecal output were not different among treatments (*p* > 0.05; Table 2).

Results for SCFAs, BCFAs, biogenic amines, phenols, and indoles are shown in Table 3 and Table 4. There was no difference in the concentrations of SCFAs and BCFAs in the feces of dogs among treatments (*p* > 0.05, Table 3). However, in the feces of animals fed 3.0YCO there were greater concentrations of putrescine and cadaverine, as well as lower histamine in comparison to control and 1.5YCO group (*p* > 0.05, Table 3). The histamine concentrations in the feces of animals fed 1.5YCO were also lower than in the control group (*p* < 0.05, Table 3). Phenol concentration was lower in the feces of dogs that consumed both diets with YCO compared to the control diet (*p* < 0.05, Table 4). However, the diets did not differ in the fecal concentration of indoles (*p* > 0.05, Table 4).

A significant reduction in fecal odor was identified in dogs supplemented with YCO (*p* < 0.05). Of the 17 evaluators, 59% (n = 10) judged the stool odor of dogs fed 1.5YCO to be less fetid compared to the control group, 18% (n = 3) to be equally fetid, and 24% (n = 4) to be more fetid. On the other hand, 100% of the evaluators (n = 17) considered the fecal odor of dogs fed 3.0YCO as less fetid than the control group.

There was greater fecal bacterial alpha diversity in the feces of dogs fed 1.5YCO and 3.0YCO, relative to the control group (*p* < 0.05, Figure 1). Furthermore, the PCoA showed evident differentiation in bacterial communities among treatments (*p* < 0.05, Figure 2). The most abundant bacterial phyla in the dogs’ feces were Bacteroidetes, Firmicutes, and Fusobacteria (Table 5). Of these, there were higher Firmicutes and lower Bacteroidetes in the feces of dogs fed 1.5YCO and 3.0YCO, relative to the control group (*p* < 0.05). It is important to note that the abundance of Bacteroidetes in the feces of dogs fed 1.5YCO was significantly lower compared to the other two diets. In addition, there was a reduction in the phylum Fusobacteria in dogs fed 3.0YCO compared to the other treatments (*p* < 0.05). A higher abundance of Actinobacteria was observed in the feces of dogs fed 1.5YCO and 3.0YCO, relative to the control group (*p* < 0.05).

There were 208 bacterial genera identified in the feces of dogs. Of these, nine differed among treatments (*p* < 0.05, Table 6). Dogs fed 1.5YCO and 3.0YCO showed a higher relative abundance of *Blautia* and *Faecalibacterium* (higher for 3.0YCO and intermediate for 1.5YCO) and a lower abundance of *Streptococcus* (*p* < 0.05) in the feces, compared to the control group. Also, dogs fed 3.0YCO showed a higher fecal relative abundance of *Turicibacter* and lower *Bacteroides* and *Fusobacterium,* compared to the other treatments (*p* < 0.05). On the other hand, dogs fed 1.5YCO showed a higher relative abundance of *Clostridium* and *Ruminococcus* and lower *Prevotella*, compared to the control group and 3.0YCO group (*p* < 0.05).

### 3.2. Experiment II: Palatability Assay

Palatability results are shown in Table 7. There was a higher number of first choices for the 3.0YCO diet, compared to the control diet (*p* < 0.05). However, the IR was higher for the diet without the additive (control) and for the 1.5YCO diet, when compared to the 3.0YCO diet (*p* < 0.001). Despite these results, the animals supplemented with YCO did not refuse any of the diets throughout the study.

## 4. Discussion

The use of functional ingredients by the pet food industry follows the growing concern of pet owners for the health and welfare of their pets. Due to the diverse properties of yeast cell wall components and EO, these become important additives to be studied and used in canine nutrition. In the present study, it potential beneficial effects of adding the YCO blend to the diet on the intestinal functionality of dogs were observed, given the modulation of the intestinal microbiota and reduction in ammonia, phenols, and histamine concentrations in feces.

The difference observed in the ATTD of DM with the 3.0YCO diet compared to the control diet contradicts previous studies that used prebiotics or EOs in dogs’ and pigs’ nutrition [1,27,28,29]. Possibly, the reduction observed in the ATTD of DM does not have a major nutritional impact, since the ATTD of other nutrients or ME were not different and a trend was only found in the digestibility of GE.

Ammonia is one of the main metabolites originating from protein fermentation by the gut microbiota [30]. When protein is not digested by host enzymes in the small intestine, gut microorganisms can hydrolyze it using extracellular proteases and peptidases, which generate free amino acids and peptides that can be absorbed by microorganisms. After the deamination process, which is the catabolic step responsible for removing the amine group from amino acids, ammonia is produced [31,32]. Therefore, the luminal ammonia concentration in the intestine can vary depending on the combined effects of microbial deamination and microbial protein synthesis [3]. A reduction in the fecal ammonia concentrations in dogs fed 3.0YCO was found in comparison to the control group. This was possibly due to the effects of YCO in controlling the growth of some proteolytic bacteria in the gut, such as *Streptococcus*.

The reduction in fecal ammonia concentrations corroborates with the result found related to fecal odor in dogs fed 3.0YCO. Ammonia, as well as biogenic amines, phenols, and BCFA, are some of the putrefactive compounds responsible for foul fecal odor [33,34]. These compounds are produced during colonic fermentation of endogenous and undigested amino acids and some of them are toxic to gut mucosal cells [35,36]. In this study, most of the evaluators judged the odor of fresh feces from dogs fed YCO as less fetid compared to the control group, in agreement with previous studies performed in dogs and cats [37,38]. Possibly, this improvement in the fecal odor occurred due to the decreased production of one or more volatile compounds from protein metabolism [37,38].

The higher fecal concentrations of some biogenic amines seem contradictory to the reduction in ammonia concentration and fecal odor in dogs fed 3.0YCO. However, it is known that some polyamines, such as putrescine and cadaverine, are extremely important for the regulation of intestinal cell physiology, such as membrane stability, correct cell proliferation, and differentiation [32,39,40,41,42,43,44]. Indeed, some studies have shown that dogs and humans with inflammatory bowel disease have reduced putrescine and spermidine concentrations [45,46]. *Faecalibacterium*, one of the bacterial genera that presented higher abundance in dogs supplemented with YCO, is able to catalyze the transfer of the propylamine group from the amine donor S-adenosylmethioninamine to putrescine, producing spermidine and increasing putrescine concentrations [46]. In addition, a relevant result was the nearly five-fold reduction in histamine concentration in the YCO-fed dogs. Histamine is an important signaling agent for toxic substances in the gut, and higher concentrations are related to intestinal inflammatory processes, such as inflammatory bowel disease, irritable bowel syndrome, and food allergy [47,48,49,50]. Thus, its reduction may be indicative of a protective effect of YCO in the gut, which may help regulate inflammatory processes.

The other fecal characteristics, such as pH, DMf, fecal score, and fecal output did not differ among treatments, contrary to Middelbos et al. [51], who identified increased fecal pH and reduced fecal output in dogs fed 0.05 to 0.65% dietary yeast cell wall. On the other hand, the study of Swanson et al. [34] demonstrated that there was no difference related to fecal characteristics in dogs fed 1 g of MOS, although a trend towards increased fecal pH was identified in animals receiving the prebiotic when compared to the control group. It is important to highlight that both studies mentioned utilized beet pulp as a fiber source, unlike the fiber composition of the present study. A possible explanation for the fact that fecal pH did not differ among treatments in the present study is the amount of YCO added to the diet. The amount added may have been insufficient to generate changes in intestinal fermentation that could be measured by fecal pH [52]. Furthermore, there is a complex interaction among fermentative metabolites produced in the gut that might or might not alter the fecal pH.

Although there was a lower mean concentration of phenols in the feces of dogs that consumed YCO diets compared to the control diet, there was no difference among the diets regarding indole production. Contrary to this result, Swanson et al. [53] identified a tendency for reduced fecal indole concentration in dogs supplemented with a mixture of oligosaccharides (MOS + FOS) and observed no difference in phenol concentration in any of the treatments.

The differences observed in the fecal concentrations of ammonia, biogenic amines, phenols, and indoles possibly occurred in response to changes in the gut microbiota. Animals fed 1.5YCO and 3.0YCO showed an increase in alpha diversity when compared to the control group. The type of diet ingested and feed additives [54,55,56], the segment of the gastrointestinal tract, and the particularities of each animal [57] are some of the factors that exert influence on the diversity of gut microbiota. MOS can modulate the gut microbiota mainly through its ability to adhere to type I fimbriae from some bacteria [58,59]. This type of fimbriae is present in most Gram-negative bacteria, such as *Escherichia coli*, *Klebsiella* sp., and some *Salmonella* sp., such as *S. typhimurium* and *S. enteritidis* [60]. Type I fimbriae allow attachment of the bacteria to the enterocyte and exert an agglutinating effect on these cells [61]. However, agglutination is blocked by D-mannose or α-methylmannosidium and by concanavalin A. Therefore, by binding with this type of fimbriae, MOS can limit intestinal colonization by potentially pathogenic microorganisms [62]. Another mechanism presented by MOS to modulate gut microbiota is through selective fermentation, which benefits the growth of certain bacterial groups, such as *Lactobacillus* and *Bifidobacterium*, and the production of SCFAs, such as acetate, propionate, and butyrate [63]. Bacteria of the genus *Lactobacillus* help maintain a proper enteric environment, as they suppress the growth of potentially pathogenic bacteria through the production of SCFAs [64]. It is important to note that this selective fermentation mechanism of the gut microbiota is secondary, as MOS is moderately fermentable [28]. Furthermore, due to the partial solubility of SCFAs in the membrane, these gut fermentation products can alter the integrity and fluidity of the membrane of pathogens, providing another mechanism for inhibiting the growth of microorganisms considered pathogenic [7].

Although there was a difference in gut bacterial diversity and richness in dogs fed YCO, fecal SCFA concentrations did not differ among treatments. Possibly, this happened because (1) SCFAs were rapidly absorbed in the gut and metabolized by the intestinal epithelium, liver, and muscle [65,66] or (2) the inclusion levels of YCO were not high enough to detect differences. Similarly, no difference in fecal SCFAs was found with the inclusion of 15 g/kg of MOS in the diet for dogs [1]. However, the inclusion of 5.23% of a prebiotic and soluble fiber blend (combination of beet pulp, FOS, inulin, MOS, and kelp) promoted increases in fecal SCFA concentrations of dogs [67].

Dogs fed 1.5YCO and 3.0YCO showed a reduction in the Bacteroidetes phylum and an increase in Firmicutes, compared to the control group, possibly as a result of the modulation of microorganisms related to intestinal eubiosis. This result contradicts the findings of Van den Abbeele et al. [68], since the authors identified a significant increase in the Bacteroidetes and Actinobacteria phyla when evaluating the effects of a *Saccharomyces cerevisiae*-based product, containing 27.5% β-glucans and 22.5% MOS, in an in vitro simulation of the canine gastrointestinal tract. On the other hand, research performed in yellow-feathered chickens showed an increase in ileal Firmicutes and a reduction in the relative abundance of ileal Proteobacteria and Actinobacteria in chickens supplemented with 150 or 300 mg/kg of oregano EO [69].

Another component of the yeast cell wall that could have had an association with the modulation of gut microbiota and intestinal functionality is the β-glucans. It is known that β-glucans exert biological effects on the organism, such as immunomodulation [70,71]. When in the host, β-glucans bind to the Dectin-1 receptor, stimulating the production of many cytokines or other mechanisms of immune and non-immune reactions [72]. Research performed in rats revealed that β-glucans can produce effects on the systemic immune system and interact with the gut-associated lymphoid tissue, modulating the expression of pattern recognition receptors in this tissue [73]. Furthermore, another positive effect of β-glucans on gastrointestinal functionality reported in mice is the regulation of gut microbiota through the production of SCFAs [74].

Beyond yeast cell wall components, EOs may be associated with the modulation of gut microbiota and intestinal functionality. The antimicrobial action of EOs and the modulation of the gut microbiota occur due to the action of bioactive compounds on several targets in the bacterial cell, such as damage to the cell membrane, cytoplasmic membrane, and protein membrane, extravasation of cell contents, coagulation of the cytoplasm, and the depletion of the proton motor force [75]. In general, EOs have a greater spectrum of action on Gram-positive bacteria than Gram-negative bacteria, since Gram-negative bacteria are more resistant to EOs [76]. However, carvacrol and thymol, recognized for having intense antimicrobial activity, also have action on Gram-negative bacteria. These phenolic compounds can disintegrate the outer membrane of Gram-negative bacteria, releasing lipopolysaccharides and increasing the permeability of the cytoplasmic membrane [77]. It is also possible that the variation in composition among EOs is sufficient to vary the degree of susceptibility of Gram-negative and Gram-positive bacteria. Due to the hydrophobicity of EO and their compounds, the membrane polysaccharides, fatty acids, and phospholipid layer of the bacterial cell wall and mitochondria are injured, generating changes in structural conformation, and making the membrane permeable [78,79].

In a study by Zeng et al. [80], when evaluating the use of an EO blend consisting of 4.5% cinnamaldehyde and 13.5% thymol in weaned pigs, animals fed EOs showed a significant reduction in *E. coli* and total anaerobic bacteria in the rectum and a quantitative increase in *Lactobacillus* in the colon and rectum, when compared to pigs that did not receive such supplementation. Extrapolating to other animal species, some studies encompass the use of phytogenic compounds in canine nutrition. In one of these [81], it was shown that dogs fed a blend composed of 21.55 mg/g carvacrol, 18.76 mg/g, thymol, and 27.62 mg/g cinnamaldehyde showed a reduction in the total bacterial count, total coliforms, *Salmonella* spp., and *E. coli*, revealing the important effect of these compounds in improving the host interaction with the gastrointestinal microbiota, which is one of the key components of intestinal functionality [82].

From the bacterial genera that increased in the feces of dogs fed the YCO, *Faecalibacterium* and *Blautia* are known as butyrate producers and are associated with a lower incidence and severity of inflammatory processes in the gut. These bacteria are considered biomarkers of gut functionality in terms of normal and stable microbiota, effective immune status, and gut mucosa [66,82,83]. Also, *Faecalibacterium prausnitzii*, the only known species of this genus, secretes metabolites that block the activation of NF-kB factor transcription, consequently resulting in the inhibition of the production of pro-inflammatory interleukins, such as interleukin 8 [84].

It is important to highlight the greater relative abundance of *Clostridium* in the feces of dogs fed 1.5YCO when compared to the control group. Although the *Clostridium* genus is recognized for having species with potential pathogenicity to animals, such as *C. difficile* and *C. perfringens* [85,86], studies reveal the beneficial effects of *C. hiranonis* in dogs by converting primary bile acids into secondary ones [66]. Secondary bile acids control the growth of *C. difficile* spores and, in previous studies in dogs, have been shown to stimulate the growth of *Faecalibacterium* and inhibit *E. coli* [87], being a mechanism for controlling the growth of potentially pathogenic microorganisms. Considering this, it is important that future studies evaluate the effects of YCO and other additives on gut bacterial species to better understand the relationships among microbial species and to aid in the development of beneficial feed additives.

The greater bacterial diversity in the feces of dogs fed YCO is one of the main findings related to improved intestinal functionality. Dogs with gastroenteritis, such as inflammatory bowel disease and acute and chronic diarrhea, have a lower diversity of the gut microbiota, characterizing dysbiosis [88]. Unlike the healthy dogs enrolled in this study, which have a higher abundance of *Faecalibacterium* and *Blautia* and a lower concentration of *Streptococcus*, several studies show that dogs with gastroenteritis have reduced concentrations of key microorganisms, such as *Faecalibacterium*, *Blautia*, and *Turicibacter*, and increased *Streptococcus* [88,89,90,91,92].

Regarding palatability, to the author’s knowledge, no studies that evaluate the palatability of diets containing oregano EO in dogs have been published. However, one study evaluating a blend of EOs (copaiba, cashew nutshell, and peppers) described a possible negative effect of the EO blend evaluated on diet palatability in dogs [13].

In this study, the inclusion of YCO resulted in lower feed consumption compared to the control diet, even though some yeast-derived products, like sugarcane yeast, are usually known to make diets more palatable [15]. Such result may have occurred due to (1) the organoleptic characteristics of oregano EO, which presents intense odor and flavor, and (2) the inclusion of the YCO blend by coating, which may have accentuated the flavor and odor. Possibly, the inclusion of the YCO blend in the dough before extrusion would have less influence on palatability and feed consumption. However, further studies are needed to confirm this hypothesis.

## 5. Conclusions

Although the inclusion of 3.0YCO reduced dietary DM digestibility and palatability, it did not alter the ATTD of other nutrients, including ME. Overall, dogs fed dry food coated with the YCO blend had indicators of improvement in intestinal functionality, characterized by greater fecal bacterial diversity, modulation of microorganisms considered beneficial, and lower histamine, phenol, and ammonia concentrations.

## Figures and Tables

**Figure 1 animals-13-02527-f001:**
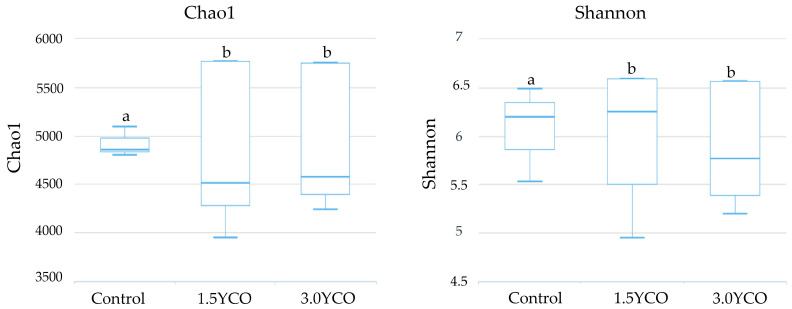
Chao1 and Shannon indexes of the fecal microbiota of dogs fed diets without (control) or with yeast cell wall and oregano essential oil (YCO). ^a,b^ Different letters indicate difference by the Dunn’s test (*p* < 0.05).

**Figure 2 animals-13-02527-f002:**
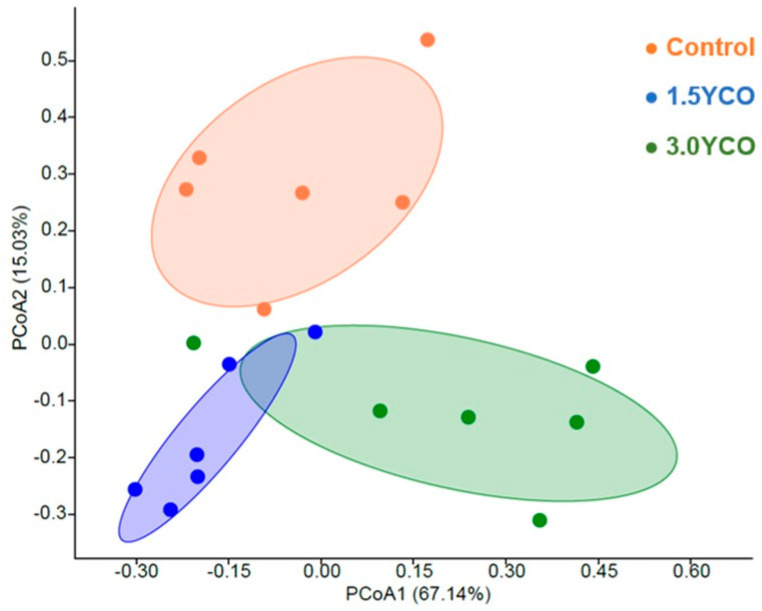
Principal coordinate analysis (PCoA) by Bray–Curtis dissimilarity method showing the different groupings of the treatments: control and 1.5 and 3.0 kg/ton of yeast cell wall and oregano essential oil (YCO). The effect of the treatments explains 67.14% of the variation in the data. The three groups differed according to the ANOSIM analysis (*p* < 0.05).

**Table 1 animals-13-02527-t001:** Analyzed chemical composition (g/kg of dry matter) of the diets without (control) and with yeast cell wall and oregano essential oil (YCO).

Item	Control	1.5YCO	3.0YCO
Dry matter	922.3	922.1	924.7
Crude protein	184.6	184.4	183.9
Ether extract in acid hydrolysis	113.7	121.4	114.9
Ash	7.43	7.18	8.23
Total dietary fiber	62.3	60.6	63.8
Insoluble fiber	45.7	44.9	47.6
Soluble fiber	16.6	15.7	16.2
Calcium	1.90	1.89	1.91
Phosphorus	1.05	1.02	1.11
Gross energy (MJ/Kg of dry matter)	19.59	19.69	19.52

**Table 2 animals-13-02527-t002:** Apparent total tract digestibility (ATTD), metabolizable energy (ME, MJ/kg), and fecal characteristics of dogs fed diets without (control) or with yeast cell wall and oregano essential oil (YCO) at two different levels of inclusion, 1.5 and 3.0 kg/ton.

Item	Control	1.5YCO	3.0YCO	SEM ^1^	*p*-Value
ATTD					
Dry matter	0.757 ^a^	0.754 ^ab^	0.735 ^b^	0.0061	0.023
Organic matter	0.786	0.789	0.770	0.0076	0.182
Crude protein	0.801	0.800	0.787	0.0066	0.271
Ether extract	0.875	0.883	0.892	0.0073	0.289
Gross energy	0.797	0.795	0.781	0.0052	0.077
ME (MJ/kg of dry matter)	15.55	15.59	15.29	0.0636	0.109
Fecal characteristics					
Dry matter (g/kg)	319.2	323.3	338.6	0.782	0.215
Production	0.73	0.76	0.78	0.017	0.488
pH	5.57	5.48	5.48	0.072	0.868
Ammonia (g/kg)	0.54 ^a^	0.54 ^a^	0.32 ^b^	0.002	0.018
Score ^2^	4.0	4.0	4.0	-	0.632

^1^ SEM = standard error of the mean. ^2^ Score: 1 = watery and unformed feces to 5 = well-formed, hard, and dry feces. Score was analyzed by Kruskal–Wallis (*p* < 0.05). ^a,b^ Means followed by different letters differ by Tukey’s test (*p* < 0.05).

**Table 3 animals-13-02527-t003:** Fecal concentrations (dry matter basis) of short-chain (SCFAs) and branched-chain (BCFAs) fatty acids and biogenic amines of dogs fed diets without (control) or with yeast cell wall and oregano essential oil (YCO) at two different levels of inclusion, 1.5 and 3.0 kg/ton.

Item	Control	1.5YCO	3.0YCO	SEM ^1^	*p*-Value
SCFAs (µmol/g)					
Acetate	35.97	31.85	32.54	1.779	0.284
Propionate	20.12	22.09	19.32	0.975	0.142
Butyrate	3.66	3.55	3.86	0.459	0.957
Total	59.75	57.48	55.72	2.383	0.537
BCFAs (µmol/g)					
Isobutyrate	0.32	0.27	0.33	0.003	0.999
Valerate	0.07	0.03	0.03	0.001	0.999
Total	0.39	0.30	0.35	0.003	0.999
Biogenic amines (mg/kg)					
Tiramine	2.16	2.67	3.70	0.709	0.199
Putrescine	30.80 ^b^	26.93 ^b^	42.52 ^a^	3.295	0.007
Cadaverine	5.96 ^b^	4.11 ^b^	25.84 ^a^	3.630	0.002
Histamine	26.53 ^a^	16.67 ^b^	5.83 ^c^	4.917	0.035
Serotonin	0.00	0.20	0.00	0.001	0.391
Espermidine	37.78	37.63	34.93	2.357	0.651
Tryptamine	0.00	0.23	0.00	0.117	0.391
Total amines	103.23	88.21	112.82	9.25	0.102

^1^ SEM = standard error of the mean. ^a,b^ Means followed by different letters differ by Tukey’s test (*p* < 0.05).

**Table 4 animals-13-02527-t004:** Percentage medians of the peak area of the most abundant volatile organic compounds present in the feces of dogs fed diets without (control) and with yeast cell wall and oregano essential oil (YCO) at two different levels of inclusion, 1.5 and 3.0 kg/ton.

Item	Control	1.5YCO	3.0YCO	*p*-Value
Phenols	0.4020 ^a^	0 ^b^	0 ^b^	0.023
Indoles	1.9980	1.2550	0.8993	0.615

^a,b^ Medians followed by different letters differ by Dunn’s test (*p* < 0.05).

**Table 5 animals-13-02527-t005:** Relative abundance (%) of the most abundant phyla in the fecal microbiota of dogs fed diets without (control) or with yeast cell wall and oregano essential oil (YCO) at two different levels of inclusion, 1.5 and 3.0 kg/ton.

Item	Control	1.5YCO	3.0YCO	SEM ^1^	*p*-Value
Actinobacteria	1.45 ^b^	4.47 ^a^	3.13 ^a^	1.752	0.489
Bacteroidetes	55.38 ^a^	31.98 ^c^	42.63 ^b^	2.701	<0.001
Firmicutes	34.15 ^b^	54.28 ^a^	48.23 ^a^	3.144	0.001
Fusobacteria	5.55 ^a^	5.89 ^a^	2.18 ^b^	0.923	0.023
Proteobacteria	2.48	2.44	1.54	0.386	0.185
Tenericutes	0.14	0.10	0.06	0.001	0.243

^1^ SEM = standard error of the mean. ^a,b^ Means followed by different letters differ by Tukey’s test (*p* < 0.05).

**Table 6 animals-13-02527-t006:** Relative abundance (%) of the most abundant genera in the fecal microbiota of dogs fed diets without (control) or with yeast cell wall and oregano essential oil (YCO) at two different levels of inclusion, 1.5 and 3 kg/ton.

Item	Control	1.5YCO	3.0YCO	SEM ^1^	*p*-Value
*Bacteroides*	12.21 ^a^	12.65 ^a^	3.66 ^b^	2.214	0.019
*Blautia*	6.05 ^b^	11.47 ^a^	10.55 ^a^	1.531	0.006
*Clostridium*	1.78 ^b^	2.72 ^a^	1.67 ^b^	0.219	0.007
*Faecalibacterium*	2.00 ^b^	3.05 ^ab^	3.90 ^a^	0.382	0.011
*Fusobacterium*	5.54 ^a^	5.88 ^a^	2.18 ^b^	0.927	0.024
*Prevotella*	36.92 ^a^	16.47 ^b^	37.09 ^a^	4.201	0.004
*Ruminococcus*	0.34 ^b^	1.53 ^a^	0.43 ^b^	0.287	0.018
*Streptococcus*	0.99 ^a^	0.01 ^b^	0.01 ^b^	0.029	0.009
*Turicibacter*	4.25 ^b^	7.39 ^b^	13.21 ^a^	1.667	0.005

^1^ SEM = standard error of the mean. ^a,b^ Means followed by different letters differ by Tukey’s test (*p* < 0.05).

**Table 7 animals-13-02527-t007:** First choice (%) and intake ratio (%) of the control and experimental diets containing yeast cell wall and oregano essential oil (YCO) at two different levels of inclusion, 1.5 and 3 kg/ton.

Item	Test 1		Test 2		Test 3	
	Control	1.5YCO	Control	3.0YCO	1.5YCO	3.0YCO
First choice ^1^	46.9	53.1	31.2	68.8 *	46.9	53.1
Intake ratio ^2^	80.4 **	19.6	74.8 **	25.2	87.3 **	12.7

^1^ First choice by Chi-square test (*p* < 0.05) ^2^ Intake ratio by paired Student’s *t*-test (*p* < 0.05). * *p* < 0.05 ** *p* < 0.01.

## Data Availability

Data sharing not applicable.

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
