# Peer review of "Digestibility and Palatability of the Diet and Intestinal Functionality of Dogs Fed a Blend of Yeast Cell Wall and Oregano Essential Oil"

_animals, 2023, doi:10.3390/ani13152527_

Round 1

Reviewer 1 Report

Dear Authors,

Congratulations on nice and interesting reasearch. Some comments below:

Line 10 and 11 - Feed additives are commonly used by many pet food industries as they can exert important properties on the animal. Omit the sentence.

Prebiotics and essential oils are some of the feed additives that can be added to the diet, individually or in combination. – please add the reason for the addition of essential oils or prebiotics in pet food diets

Line 19 The combined effects of yeast cell wall….. were analyzed.

Lines 120 and 121 Experimental design should be explained a bit clearer as it seems to be a mistake in the description, or it is unclear.

Line 134 separately by animal – omit

Line 325-336 Names of bacteria should be in italic

The quality of English is good, but the paper would benefit from a review by a native speaker.

Author Response

We thank very much the reviewer for all the suggestions. 

Reviewer 2 Report

Well written manuscript evaluating the inclusion at 2 levels of a combination product containing yeast cell wall and oregano. While a study evaluating the effects of the two separately would be ideal, this design, and presentation of data were well designed.  I believe a follow up study might be to look at each individually to determine if one or the other might have an impact.

Well written manuscript with significant data presented.  I would suggest very minor edits:

Line 69: Change to "aim of the study was to"

Line 163: Do the authors mean "smelt"?

Results: It would be more helpful to the reader if a reference to the respective tables was highlighted initially. The presentation is currently consistent with table references at the end of each text section but it might be easier to read if they were placed first.

Author Response

(The authors gave the same response as above.)

Reviewer 3 Report

The paper entitled “Digestibility and palatability of the diet and intestinal functionality of dogs fed a blend of yeast cell wall and oregano essential oil” describes an experimental study (randomized block design) addressing nutritional claims of prebiotic feed additives at two different levels for complete dry pet foods. Comparisons were made against an identical control diet that differ only in the inclusion of the feed additives being tested. Authors evaluated several parameters in dog’s feces including score, odor, metabolites, microbial composition, and diversity, as well as nutrient digestibility and palatability of diets. The topic is relevant, and the study is well designed, although it is my understanding that some methods need to be clarified. Also, the conclusion made by the authors is that the inclusion of feed additives is beneficial for dogs, which I did not think is fully supported by the results herein reported. The discussion section is maybe too extensive: it includes an interesting revision of the literature but is not always linked to the results, which are insufficiently discussed. I provide specific comments:

L 37 – Additives can be ingredients with prebiotics properties.

L 37 and elsewhere - I suggest replacing "functionality" with "function" on the whole document.

L 51 – 55 - I think it is worth mentioning if these effects have been reported on dogs or other species.

L 88 to 89 - This means leash walks or outdoor parks where all dogs interact?

L 90 to 91 - Is this the usual housing or just during the trials?

L 94 - This is high maintenance; do dogs exercise a lot? I think housing and routine should be better described. Stress-related to the study (assuming that routines change) can induce significant modifications in the microbiota (sometimes even accompanied by softer feces).

L 111 to 115 - Please include quantities of each ingredient and additive (a table maybe).

L 117 - Please provide further details on the experimental design. I see you are blocked by sex. What else? weight? age? BCS? What about the non-Beagle dogs? I suggest including a description of the adaptation and experimental period here. Also, I scheme could be useful.

L 133 - What repetition?

L 135 – A space is missing; also this is not the correct symbol for Celsius (change is needed throughout the whole document)

L 140 – MM?

L 150 – Ammonia: please provide a little description of the method.

L 152 – Is this the description of the above-mentioned "fecal production score"?

L 160 - Is this a panel of trained people? Please describe further.

L 162 to 163 - "Better" or "worse" than the control is a highly subjective analysis, I wonder if this should be included at all...

L 167 to 168 - Reference of methods is missing for all these parameters.

L 182 - Please provide a little description of the method.

L 185 - It is not clear which fecal samples were collected for this purpose. Please further describe the period of collection and the procedure related to collection and storage.

L 192 - Qiime 2?

L 201 - Why not ASV instead of OTU?

L 211 – ATTD has not been previously defined.

L 216 and elsewhere - Is “block” missing?

L 236 - I wonder if an effect of the previous feeding could have played a part... I mean preference or aversion for the diet dogs were fed during the experiment I.

L 250 to 251 - Were ever leftovers? Have dogs been fed before the test? Or exercise? Or interact with humans/ played? Have time for a decision been considered? Also, I suggest inserting an equation as this is not clear.

L 291 and elsewhere - I suggest adding "at two different levels of inclusion, 1.5 and 3 kg/ton” in the subtitles.

Table 2 - Is this NH3-N?

L 292 - Is this production in fecal DM?

L 393 -A trend was found in the digestibility of GE and close to that in ME.

L 395 - Although protein digestibility was not different... Also, differences in proteolytic bacteria are not in accordance with these results. The fecal concentration of metabolites is due to increased production or decreased absorption.

L 408 to 409 - This is not in accordance with the results though.

L 422 to 423 - Results showed that most are on amines concentration, but the magnitude of differences might not be enough to alter pH; also, there is a complexity of metabolites that might or might not have been affected by the treatment, that was not evaluated...

L437 to 438 - Can you substantiate this?

L 441 to 443 - The development/ link to your results is missing.

L 472 - Can you substantiate this?

L 473 to 474 – The balance between production and absorption of these compounds may explain no differences among treatments, which does not mean there were non; and many studies have shown differences in fecal concentration of SFCA associated with dietary prebiotics.

L 477 to 478 - This has been said in the introduction.

L 479 - It had been important to measure immunological parameters; you haven't so I think most of this paragraph can be excluded.

L 517 to 519 - Could the differences be related to the effect of EO? This likely needs to be further discussed.

L 522 - Might be higher due to MOS?

L 534 –Not herein observed for Fusobacterium

L 536 to 538 - I think the curious finding is that increase only occurred with 1.5YCO.

L 541 - Can you elaborate on this corroboration?

L 543 - Do you mean endogenous histamine? Because fecal histamine levels might indicate detoxification problems - histamine intolerance.

L 547 to 554 - I do not see the link between this and what has been said before in this paragraph.

L 570 - I do not think the results herein fully support this conclusion. Also, do the benefits of additive inclusion compensate the costs involved (costs for production will reflect on tutors’)?

Author Response

(The authors gave the same response as above.)

Reviewer 4 Report

General comments

Simple summary – instead of listing some of the nitrogen fermentative products, could you add what was the effect of these additives on digestibility and palatability? These are the first parameters you mentioned on the title and they are not mentioned here.

Abstract – somewhat similar to my comment on the Simple Summary, in the first sentence you should also include digestibility and palatability. My suggestion is to review the sentence to something like “Food additives, such as prebiotics and essential oils, are used in pet foods and can affect digestibility, palatability, and intestinal functionality of pets”. This is a mote generic sentence that encompasses your work. You did not present any results from the digestibility and palatability in the Abstract, please include these results as well.

Keywords – please add more words related to digestibility and palatability. Adding more words would help your research to be selected on search engines and be cited. I suggest you change “additives” to “feed additives”

Introduction – you use several references in the introduction that was work done with other species than dogs. You must make clear to the reader the targeted species of the cited work. This also helps you to make clear the lack of literature regarding dogs and the intent of your work in starting to fill the gap in the literature. This issue is something that you should make clear throughout the manuscript and not only in the introduction, so please revise the whole paper. I have a couple examples for you listed in my specific comments, but please revise throughout the manuscript. Overall, you focus a great deal on bacterial modulation and fermentation products and do not mention any effects of these additives in digestibility and palatability. Why are the authors avoiding writing about digestibility and palatability? Seems odd to me that since the Simple Summary the authors did not mention anything about the effects of essential oils and prebiotics on digestibility and palatability. Please revise your introduction to reflect the work you presented.

Dietary treatment abbreviations – You need to be consistent about how you will abbreviate the treatments. Under session 2.1.2 you wrote that the abbreviation would be 1.5YCO and 3.0YCO; however, this is not consistently used throughout the paper. Please refer to all tables and figures for example. There are other places in the text that you don’t use this abbreviation as well. Please, define 1 abbreviation for each of the 3 dietary treatments and only use these abbreviations in the manuscript.

Abbreviations throughout the paper – please only use abbreviations when you will use it at least once per page, otherwise they have no reason to be used. For example, you abbreviated “gut-associated lymphoid tissue” as GALT and never used the abbreviation in the paper. Please, revise the entire paper and remove any abbreviations that aren’t used at least once per page and are not commonly abbreviated terms (dry matter – DM, crude protein – CP, etc.).

Specific comments

L22-23 please review sentence as “zero, 1.5, and 3.0 kg/ton of yeast cell wall and oregano essential oil (YCO) bend”

L53 reference 8 is a literature review of research that was not done in dogs, please make sure you highlight this information. Not because those benefits were reported for other species means that dogs will have them as well. Please make sure you make these distinctions and highlight throughout the manuscript what species you are referring to if the research cited was not done with dogs.

L61 same as my comment for L53, this reference is for pigs, not dogs. It is important to make clear what was the targeted species on the work cited. Specifying the targeted species also will help to make the case for your manuscript, that there is very little work published for dogs, and that is the gap in the literature that you are trying to fill with your work.

L69 “the aim of this study was to evaluate” not “is”

L92-94 did you have to adjust food intake for any of the dogs so it would maintain body weight during the trial? If yes, please, describe how the adjustment in food allowance was made. 10% increments? How often weight was measured and food intake adjusted?

L101-102 …” and B-glucans was 5.83g/kg.”

L102 did you replace the calcium carbonate in the formula by the YCO? If yes, please change “vehicle” by “place holder”

L104 please include details about what commercial diet was used. You need to add the diet information as well as the manufacturer and lot of the diet used.

Table 1 please change “Ether extract in acid hydrolysis” by “Ether extract by acid hydrolysis”. It would be beneficial to the paper if you could add total dietary fiber, insoluble fiber, soluble fiber, B-glucan, and MOS values for the experimental diets. Crude fiber is not relevant for this experiment because this method of determining fiber does not recover soluble fractions of fiber that would help to explain the results you had for fermentation products and bacteria population changes.

L138-143 you mention twice crude protein analysis in this sentence, please revise.

L144-145 organic matter was “100% dry matter - % ash” or “100% as is - % ash”? please specify

L152-155 the description of the fecal scores does not match how fecal score was evaluated by the cited reference, please revise.

L163-165 From your description of how the odor test was set up, the samples were not blinded for the participants of the test. This is because the samples from control food were always presented first. Please explain in more detail how the test was performed and if and how the participants were blinded from the samples.

L166-168 how soon after defecation were the fecal samples collected? Because you measured volatile compounds, this information is needed.

L174 eppendorf is a brand of microcentrifuge tubes. Please revise.

L186 please remove “moment of”

L290 please rephrase as “fecal characteristics of dogs fed diets without (Control)…”

Table 2 – please add the SEM value for Ammonia. Please describe better how “Production” was estimated in the Materials and Methods session. You should not present new equations in the results or discussion sessions of your paper. Were the grams of feces produced used in the estimation of Production on a dry matter basis for the calculation or it was used on a as-is basis? Again, please provide more details on the Materials and Methods session how this variable was estimated.

L295-300 I think this paragraph should come after the reported data from table 4. It is strange that you started reporting results from table 2 in L285-286, presented the table, and then you mentioned something not related to the table, then got back to reporting results from table 2. For me moving these results after the phenol and indole data would make more sense. Or even after paragraph L301-306, but where it is now it is odd.

L388-389 the “reduction of fecal nitrogenous compounds” is not necessarily true. If you add the different biogenic amines that were analyzed, you will notice that the 3.0 inclusion level had higher total concentration of biogenic amines than the control and 1.5. I am not sure if this difference is statistically relevant. Please add the stats for the total biogenic amines on table 3 and rephrase this statement. Please also ensure that this is not repeated on the abstract and simple summary.

L390-394 I think these results are rather expected. Prebiotics are dietary compounds that are not digested by the animal’s digestive enzymes. Therefore, it is expected that more dry matter is reaching the large intestine, and if this DM is not fermented, it will increase DM excretion, thus decreasing DM digestibility. If the DM is fermentable, then there will be more bacterial growth and more bacteria in the feces. This would also help to increase the DM in the feces and decrease DM digestibility. This is one of the reasons analyzing these 3 foods for TDF, insoluble and soluble fibers is important. These analyses would provide a better understanding of the fiber composition of the foods and how they can influence DM and organic matter digestibilities/ Please rephrase this sentence.

L401-402 I don’t think enterocyte metabolism plays a role in the ammonia concentration in the large intestine. Protein digestibility is a characteristic of the protein source and not the metabolism of the enterocyte. Please revise.

L404-406 I don’t think it is as simple as it is stated here. The protein content of the food was exactly the same, and the same sources. So, the reduction in ammonia is much more likely because of the yeast and oregano essential oil stimulating bacterial growth and use the ammonia from microbial amino acid degradation to generate more microbial protein. Please revise this sentence.

L411 “gut mucosal cells” please revise

L407-414 I think you need to add more information here about these compounds and how they affect the smell of the feces. For example, if you add all the biogenic amines from 3.0YCO you have a higher number than control and 1.5YCO. Thus, not all compounds smell the same or have the same potency. It would be important to distinguish them and their effects.

L415-423 how the concentration of the yeast cell wall and MOS in the cited literature compares to your work? This might help to explain the differences in results here. Not only that, but the fiber composition of the different diets. If the cited literature diets had more fermentable fibers than your diets, this would stimulate more bacterial fermentation, etc. Please revise

L424-438 this paragraph is out of place here, it should follow paragraph L407-414. I had a similar comment previously. You need to present the data in a more linear and logic manner instead of jumping back and forth on the data you have.

L445-451 this paragraph is very similar with your next paragraph, please combine them and remove redundancies.

L473-475 not necessarily, the lack of difference between treatments could be because the inclusion levels of the yeast were not high enough to reach the limit to have a difference. Please refer back to some of the papers you cited and compare the inclusion levels and revise this sentence.

L486 what do you mean by “strengthening it”? what specifically are the parameters that are enhanced by the ingestion of B-glucans?

L513 what do you mean by intestinal functionality? Please be specific

L514-520 this paragraph is out of place here. Please move it after L475

Minor editing is needed. Please refer to my specific comments for some examples.

Author Response

(The authors gave the same response as above.)
